# Refined Composite Multiscale Fluctuation Dispersion Entropy and Supervised Manifold Mapping for Planetary Gearbox Fault Diagnosis

Haocheng Su †, Zhenya Wang †, Yuxiang Cai, Jiaxin Ding, Xinglong Wang and Ligang Yao *

School of Mechanical Engineering and Automation, Fuzhou University, Fuzhou 350108, China
* Correspondence: ylgyao@fzu.edu.cn
† These authors contributed equally to this work.

**Abstract:** A novel fault diagnosis scheme was developed to address the difficulty of feature extraction for planetary gearboxes using refined composite multiscale fluctuation dispersion entropy (RCMFDE) and supervised manifold mapping. The RCMFDE was first utilized in this scheme to fully mine fault features from planetary gearbox signals under multiple scales. Subsequently, as a supervised manifold mapping method, supervised isometric mapping (S-Iso) was applied to decrease the dimensions of the original features and remove redundant information. Lastly, the marine predator algorithm-based support vector machine (MPA-SVM) classifier was employed to achieve intelligent fault diagnosis of planetary gearboxes. The suggested RCMFDE combines the composite coarse-grained construction and refined computing technology, overcoming unstable and invalid entropy in the traditional multiscale fluctuation dispersion entropy. Simulation experiments and fault diagnosis experiments from a real planetary gearbox drive system show that the complexity measure capability and feature extraction effectiveness of the proposed RCMFDE outperform the multiscale fluctuation dispersion entropy (MFDE) and multi-scale permutation entropy (MPE). The S-Iso's visualization results and dimensionality reduction performance are better than principal components analysis (PCA), linear discriminant analysis (LDA), and isometric mapping (Isomap). Moreover, the suggested fault diagnosis scheme has an accuracy rate of 100% in identifying bearing and gear defects in planetary gearboxes.

**Keywords:** multiscale fluctuation dispersion entropy; supervised isometric mapping; feature extraction; planetary gearbox

## 1. Introduction

Planetary gearboxes are frequently employed in industrial, medical, and aerospace machinery because of their small size, strong load-bearing capacity, and huge transmission ratio. However, due to the intricate functioning and working environment, its internal parts (such as bearings and gears) are prone to malfunction. If these issues are not resolved quickly, the faulty parts could compromise the equipment's ability to function normally and result in serious mishaps. Thus, it is significant to carry out planetary gearbox fault diagnosis [1–5].

Signal collection, feature extraction, and fault detection are the three main processes in planetary gearbox failure diagnostics. Within them, feature extraction is the most crucial, which controls the ensuing diagnostic efficacy [6–9]. Fortunately, advanced nonlinear feature extraction techniques, such as entropy theories, are frequently used to extract mechanical fault characteristics. The generalized refined composite multiscale sample entropy was introduced by Wang et al. [10,11] to efficiently extract information on bearing faults. Zheng et al. validated the benefits of the defect diagnostic system through experimental data using composite multiscale fuzzy entropy [12]. Ye et al. used multi-scale permutation entropy to construct the fault characteristics of suspension systems for high-speed

trains [13]. Although the abovementioned entropy-based fault diagnosis methods have achieved good experimental results in various fields, applying these theories to planetary gearbox fault diagnosis still suffers from the following drawbacks. Specifically, computation of the generalized refined composite multiscale sample entropy is inefficient and generates uncertain values. The composite multiscale fuzzy entropy suffers from entropy instability on large scales. The multiscale permutation entropy only focuses on the alignment order but ignores the signal amplitude differences [14,15].

Fortunately, the shortcomings of the techniques above can be successfully solved by multiscale dispersion entropy, a recently created tool for nonlinear dynamics research. Moreover, the multi-scale dispersion entropy offers the benefits of quick calculation and noise immunity [16]. Azami et al. developed the multiscale fluctuation dispersion entropy (MFDE) with a new diffusion graph fluctuation evaluation approach to enhance the analytical performance of the original multiscale dispersion entropy method [17]. However, MFDE application to planetary gearbox feature extraction is still characterized by shortcomings. The MFDE uses the traditional coarse-grained construction technique, resulting in unstable analysis results at large scales. A novel and refined composite multiscale fluctuation dispersion entropy (RCMFDE) method is proposed in this paper and applied to address this defect and fully extract planetary gearbox features.

However, the RCMFDE-based fault feature extraction has redundant fault features, which might lengthen detection time and reduce detection precision if immediately input to a classifier for fault detection. Hence, it is essential to utilize dimensionality reduction techniques for effective secondary feature extraction [18,19]. Principal component analysis (PCA) and linear discriminant analysis (LDA) are the major types of conventional dimensionality reduction techniques. However, both are linear dimensionality reduction techniques [20–23], indicating that the abovementioned methods are unsuitable for dealing with nonlinear fault feature sets of planetary gearboxes [24]. Subsequently, several nonlinear manifold learning techniques for fault feature fusion have been proposed, including isometric mapping (Isomap), linear local tangent space alignment, and discriminant diffusion mapping analysis [25–27]. However, such nonlinear methods are unsupervised dimensionality reduction techniques that do not fully utilize dataset label information. Therefore, in this paper, a novel supervised Isomap (S-Iso) method was applied to achieve dimensionality reduction of the RCMFDE feature sets [28].

In summary, a unique planetary gearbox feature extraction method predicated on RCMFDE and S-Iso was established in this paper to obtain the feature matrices with the ability to distinguish faults easily. Moreover, the marine predator algorithm-based support vector machine (MPA-SVM) was employed as a fault identifier for achieving the intelligent diagnosis of planetary gearboxes [29–31]. Simulations and planetary gearbox fault diagnosis experiments confirm the efficiency and superiority of the suggested methods. The contributions of this paper are as follows:

1. A novel RCMFDE method is developed to extract planetary gearbox fault features.
2. An entropy-manifold-based feature extraction technique is proposed to mine the sensitive components by combining the RCMFDE and S-Iso.
3. A fault detection scheme for planetary gearboxes is developed based on the entropy-manifold characteristics and MPA-SVM.
4. Planetary gearbox fault diagnosis experiments are carried out to assess the viability of the suggested approaches. The results show that the RCMFDE outperforms the existing MFDE [17] and multi-scale permutation entropy (MPE) [8] for feature extraction. The dimensionality reduction of the S-Iso is better than the well-established PCA [22], LDA [21], and Isomap [25]. The suggested scheme can accurately determine various planetary gearbox faults.

The remainder of this paper is organized as follows. Section 2 provides the specific procedures of the proposed RCMFDE and conducts simulation experiments to verify its effectiveness. Section 3 outlines the suggested planetary gearbox fault diagnosis scheme,

while the fault diagnosis experiments on a real planetary gearbox are investigated in Section 4. Ultimately, Section 5 summarizes a few key findings.

## 2. Refined Composite Multiscale Fluctuation Dispersion Entropy Method

*2.1. Multiscale Fluctuation Dispersion Entropy*

Given a time series $Y = \{y_1, y_2, \ldots, y_N\}$, the MFDE's fundamental procedure is as follows.

5.   The new coarse-grained sequences $Z^{(s)} = \{z_1, z_2, \ldots, z_{N/s}\}$ are constructed by:

$$z_j = \frac{1}{s} \sum_{i=(j-1)s+1}^{js} y_i \tag{1}$$

where $s$ is the scaling factor, and $z_j$ denotes the value of the $j$-th new sequence.

6.   The $Z^{(s)}$ are mapped to between 1 and $c$.

$$x_i = \frac{1}{\sigma\sqrt{2\pi}} \int_{-\infty}^{z_i} \exp[-(h-\mu)^2/2\sigma^2]\, dh \tag{2}$$

$$W_i = R(c \cdot x_i + 0.5) \tag{3}$$

where $\mu$ is the expectation, $\sigma^2$ is the variance, $c$ is the category, $W = \{W_i\}$ represents the mapping result, and $R$ is the rounding function.

7.   The mapping result $W$ is reconstructed by:

$$U_j^{(m,\,t,c)} = \left\{ W_j,\, W_{j+t}, \ldots,\, W_{j+(m-1)t} \right\},\ j = [1,\, 2, \ldots,\, N - (m-1)\, t] \tag{4}$$

where $m$ is the embedding dimension and $t$ is the time delay. $U_j^{(m,\,t,c)}$ is the $j$-th reconstructed vector.

8.   The reconstruction result $\mathbf{U}^{(m,\,t,c)} = \left\{ U^{(m,\,t,c)} \right\}$ is transformed to the fluctuation dispersion result $\mathbf{Q}^{(m,\,t,c)} = \left\{ Q_j^{(m,\,t,c)} \right\}$.

$$Q_j^{(m,\,t,c)} = \left\{ W_{j+t} - W_j, \ldots,\, W_{j+(m-1)t} - W_{j+(m-2)t} \right\} \tag{5}$$

where $Q_j^{(m,\,t,c)}$ denotes the $j$-th fluctuation dispersion.

There is a specific fluctuation dispersion pattern for each time series: $\Omega_{v_0 v_1 \ldots v_{m-2}}$ ($1 \leq v \leq 2c - 1$), $W_{j+t} - W_j = v_0$, ..., $W_{j+(m-1)t} - W_{j+(m-2)t} = v_{m-1}$. Then, every pattern's probability $p(\Omega_{v_0 v_1 \ldots v_{m-2}})$ is determined by:

$$p(\Omega_{v_0 v_1 \ldots v_{m-2}}) = \frac{Number(\Omega_{v_0 v_1 \ldots v_{m-2}})}{N - (m-1)\, t} \tag{6}$$

9.   The entropy values of the MFDE are computed by:

$$MFDE(Y, m, c, t, s) = -\sum_{\Omega=1}^{(2c-1)^{m-1}} p(\Omega_{v_0 v_1 \ldots v_{m-2}}) \ln p(\Omega_{v_0 v_1 \ldots v_{m-2}}) \tag{7}$$

*2.2. Refined Composite Multiscale Fluctuation Dispersion Entropy*

Using the MFDE to extract nonlinear features of planetary gearboxes may result in unstable entropy values and insufficient excavation of signal information. Therefore, the RCMFDE was developed. Figure 1 depicts the RCMFDE flowchart, and the specific stages are as follows:

10.   The composite coarse-grained technique with Equation (8) was used instead of the traditional one. A comparison of these two methods is displayed in Figure 2. Com-

pared with the conventional coarse-grained technique, the composite coarse-grained technique can more fully exploit useful information from the time series.

$$q_{k,n} = \frac{1}{s}\sum_{i=n+(k-1)s}^{n+ks-1} y_i, \; 1 < n < s, \; 1 < k < N/s \tag{8}$$

11. The fluctuation dispersion pattern probabilities for each new sequence were calculated according to steps 2–4 in the MFDE.
12. The entropy values of the RCMFDE were computed using the refined arithmetic approach:

$$RCMFDE(Y,m,c,t,s) = -\sum_{\Omega=1}^{(2c-1)^{m-1}} \overline{p}\left(\Omega_{v_0 v_1 \ldots v_{m-2}}\right) ln \overline{p}\left(\Omega_{v_0 v_1 \ldots v_{m-2}}\right) \tag{9}$$

where $\overline{p}(\Omega_{v_0 v_1 \ldots v_{m-2}})$ is the mean probability that each coarse-grained sequence will have a fluctuation dispersion pattern.

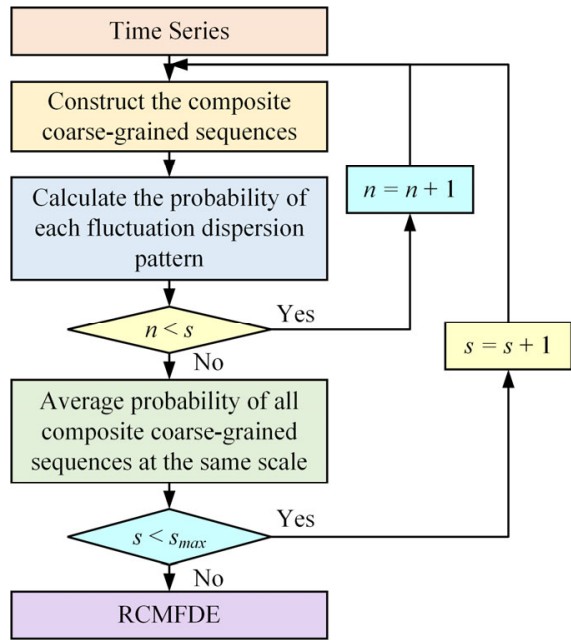

**Figure 1.** Flowchart of refined composite multiscale fluctuation dispersion entropy (RCMFDE).

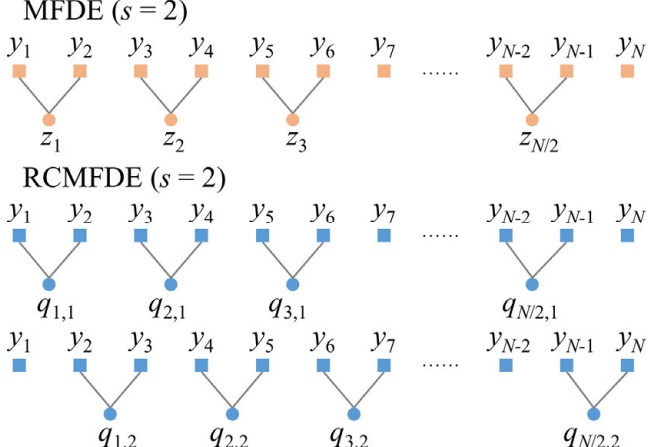

**Figure 2.** Comparison of two coarse-grained techniques.

The invalid entropy value can appear only when the probability of all coarse-grained sequences is zero. Therefore, this refined operation can reduce the possibility of invalid entropy values.

### 2.3. Simulation Experiments

Fifty sets of white noise signals and 50 sets of blue noise signals were utilized in the experiment to examine the impact of each parameter (i.e., signal length $N$, embedding dimension $m$, category $c$, time delay $t$, and scale factor $s$) on the RCMFDE method.

14. The effect of the scale factor $s$ on the performance of the RCMFDE is investigated. When $s$ is too low, the RCMFDE cannot fully exploit the entropy information of the signal. Conversely, when $s$ is excessive, the RCMFDE is prone to invalid or inaccurate entropy results at large scales. According to [17,24], the performance $s$ is defined as 25 in the paper.

15. The effectiveness of signal length $N$ on the performance of RCMFDE is investigated. The complexity analysis is performed using RCMFDE for noise signals with different lengths (set to 1000, 2000, 3000, and 4000), and the entropy curves are plotted in Figure 3. In this experiment, $m = 2$, $c = 6$, $t = 1$, and $s = 25$.

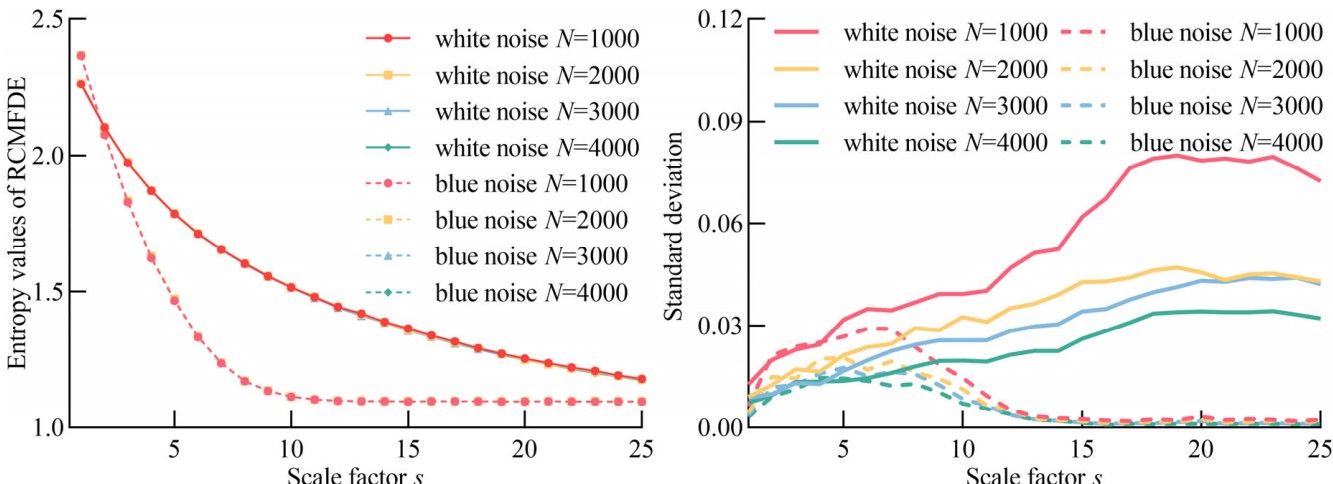

**Figure 3.** Effect of parameter $N$ on RCMFDE.

The following conclusions are obtained from Figure 3.

The mean entropy curves of the same noise with different $N$ are close to each other, indicating that the parameter $N$ has less influence on the analysis results of the RCMFDE. However, the standard deviation of the RCMFDE entropy values decreases with an increase in $N$, indicating that increasing the signal length can improve algorithm stability. Besides, the RCMFD requires more running time with an increase in $N$ (according to Table 1). Thus, the parameter $N$ is set to 3000 in this paper by considering the stability and operation efficiency.

**Table 1.** Running time comparison of RCMFDE different $N$.

| Noise | $N = 1000$ | $N = 2000$ | $N = 3000$ | $N = 4000$ |
|---|---|---|---|---|
| White noise | 2.71 s | 4.58 s | 6.45 s | 8.21 s |
| Blue noise | 2.75 s | 4.55 s | 6.41 s | 8.31 s |

16. The effect of the embedding dimension $m$ on RCMFDE performance was investigated. The complexity analysis was performed using RCMFDE for white noise signals with different values of $m$ (set to 2, 3, 4, and 5), and the analysis results are provided in Figure 4. In this experiment, $N = 3000$, $c = 6$, $t = 1$, and $s = 25$.

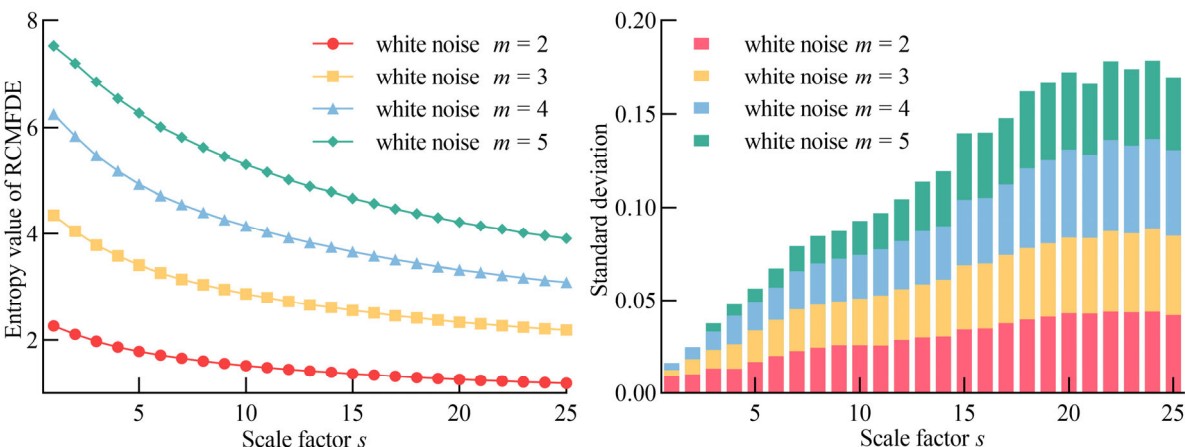

**Figure 4.** Effect of parameter m on RCMFDE.

Figure 4 demonstrates that the white noise exhibits higher mean entropy outcomes with an increase in $m$. This is because the fluctuation dispersion pattern of RCMFDE increases with $m$, increasing the entropy analysis results. Moreover, the inclination of the average entropy curve of the white noise is approximately the same for different values of $m$. However, as $m$ increases, the standard deviation of the entropy values rises. Therefore, the parameter $m$ is set to 2.

17. The effect of category $c$ on the performance of RCMFDE is investigated. The complexity analysis is performed using RCMFDE for white noise signals with different values of $c$ (set to 5, 6, 7, and 8), and Figure 5 displays the analysis findings. The setup values for the remaining parameters are $N = 3000$, $m = 2$, $t = 1$, and $s = 25$.

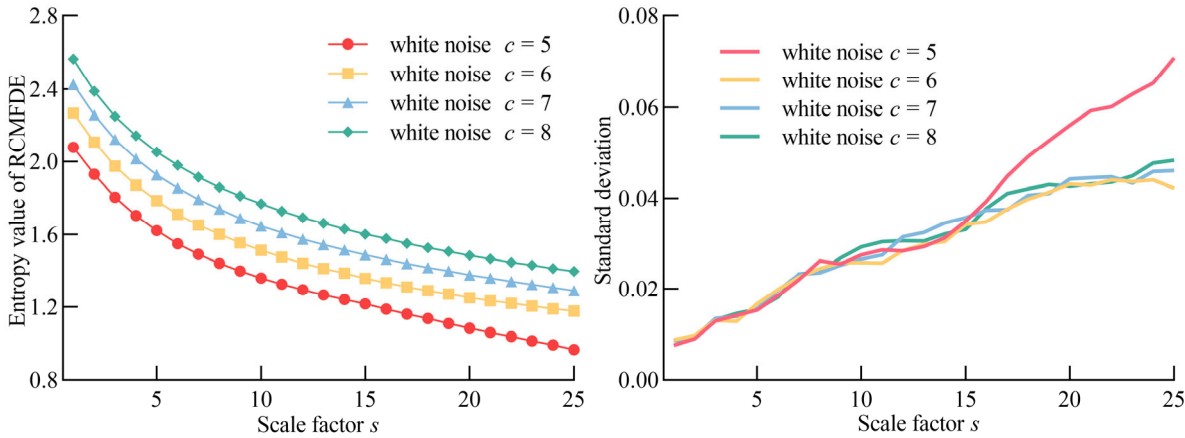

**Figure 5.** Effect of parameter c on RCMFDE.

According to Figure 5, the mean entropy curves increase with $c$. This is because the fluctuation dispersion patterns increase with $t$, resulting in larger entropy values. Secondly, the tendencies of the mean RCMFDE entropy curves for the white noise are approximately the same for different values of $c$. However, high standard deviation values appear on large scales when parameter $c$ is smaller (i.e., 5), while the standard deviation curves are closer when $c$ is 6–8. Therefore, $c$ was set to 6 in this paper.

18. The investigation is done into how the time delay $t$ affects how well the RCMFDE performs. The complexity analysis was performed using RCMFDE for white noise signals with different $t$ (set to 1, 2, 3, and 4), and the analysis results are displayed in Figure 6. In the experiment, $N = 3000$, $m = 2$, $c = 6$, and $s = 25$.

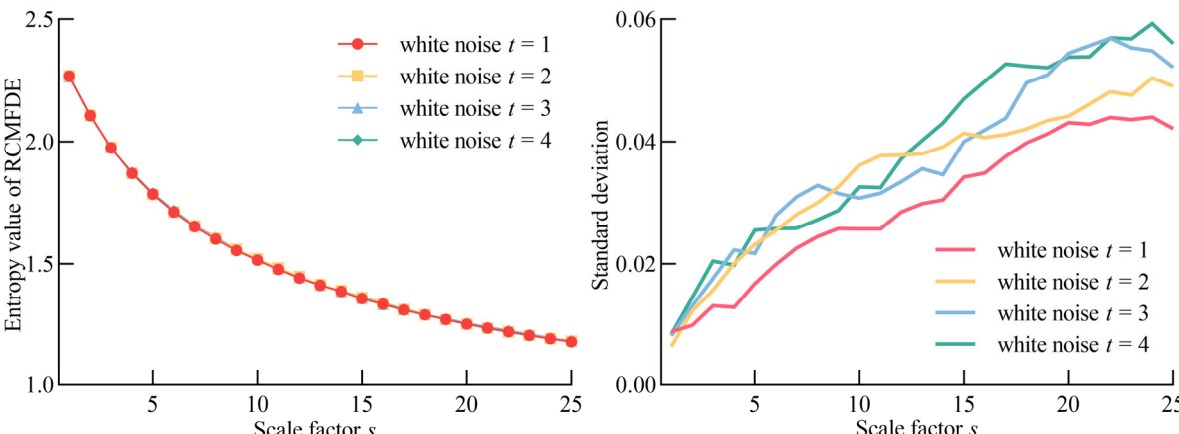

**Figure 6.** Effect of parameter *t* on RCMFDE.

According to Figure 6, the mean RCMFDE entropy curves of the white noise are nearly the same at different values of *t*. However, as *t* increases, the standard deviation of the entropy values rises. Therefore, the parameter *t* is set to 1.

To summarize, the key parameters of the RCMFDE are set as follows: $N = 3000$, $m = 2$, $c = 6$, $t = 1$, and $s = 25$.

Further, the suggested RCMFDE approach for feature extraction is compared against MFDE [17] and MPE [8] approaches. Figure 7 illustrates the analysis outcomes for these three approaches for two noise signals.

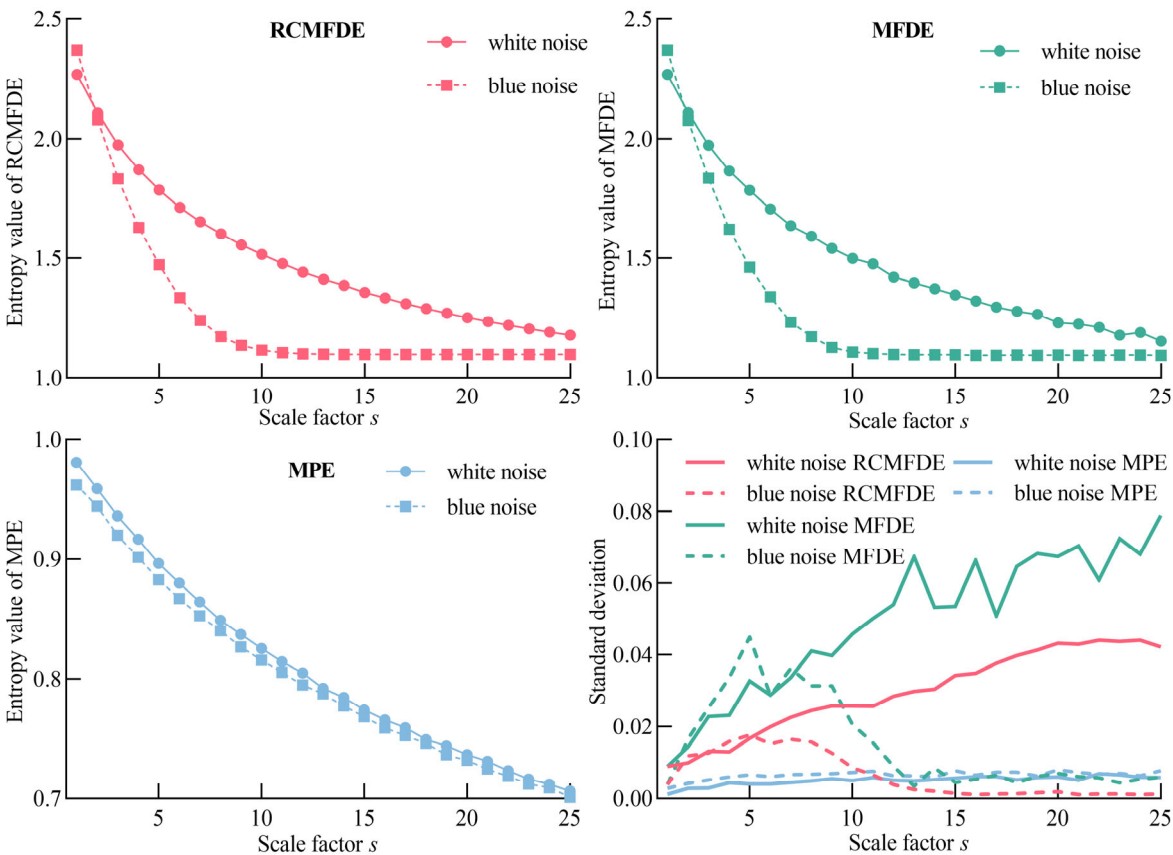

**Figure 7.** Comparison results of different entropy theories.

The following conclusions can be obtained from Figure 7.

First, the differentiation of these two noises by the MPE is not better than that by MFDE and RCMFDE, indicating that MFDE and RCMFDE can mine the useful feature information with distinguishability.

Second, compared with the RCMFDE, the average MFDE entropy curves have certain fluctuations on enormous scales with higher standard deviations, confirming that the RCMFDE with the refined composite idea can improve entropy instability and bias of the original MFDE. Moreover, the RCMFDE can obtain more stable feature extraction results. The above analyses demonstrate the potency and superiority of the suggested RCMFDE method.

## 3. Planetary Gearbox Fault Diagnosis Scheme

### 3.1. Supervised Isometric Mapping

The fault feature set extracted by RCMFDE has a redundancy that affects the subsequent classifier recognition. Consequently, the dimensionality reduction approach should be used to perform the second feature extraction. The S-Iso combines supervised learning theory with nonlinear manifold learning, showing good dimensionality reduction performance. Thus, it was used to reduce the fault feature set's dimensionality. Given a feature set $A = [a_1, a_2, ..., a_n]$, the following is a summary of the S-Iso steps:

19. The supervisory distance matrix $D_s = \{d_s(a_i, a_j)\}$ is constructed as follows:

$$d_s(a_i, a_j) = \begin{cases} \sqrt{1 - \exp[(-d^2(a_i, a_j))/\psi]}, & L(a_i) = L(a_j) \\ \sqrt{\exp[(d^2(a_i, a_j))/\psi] - \lambda}, & L(a_i) \neq L(a_j) \end{cases} \tag{10}$$

where $d(a_i, a_j)$ is the Euclidean distance between points $a_i$ and $a_j$, $L(a_i)$ is the label of $a_i$, $L(a_j)$ is the label of $a_j$, $\psi$ is the whole sample pairings' average Euclidean distance, and $\lambda$ is the weight parameter.

20. The neighborhood graph was created using the K-nearest neighbor algorithm. If $a_i$ is the nearest neighbor point of $a_j$, there exists an edge connection with length $d_s(a_i, a_j)$; otherwise, the edge length between these two points is infinite.

21. The geodesic distance matrix $D_g = \{d_g(a_i, a_j)\}$ was approximated using Dijkstra's algorithm:

$$d_g(a_i, a_j) = \min\{d_s(a_i, a_j), d_s(a_i, a_l) + d_s(a_i, a_l)\} \tag{11}$$

where min{ } is the smallest value in the set.

22. The dimensionality reduction result of $D_\mathbf{g}$ can be obtained by multidimensional scaling.

### 3.2. Marine Predators Algorithm-based Support Vector Machine

The recognition performance of the SVM classifier is vulnerable to the penalty parameter $c$ and the kernel parameter $g$. The advanced marine predators' algorithm (MPA) was employed to adjust the parameters and overcome this shortcoming.

23. The input training and testing sets are normalized. The initialized population is 20, the maximum number of iterations $T_{\max}$ is 100, and the predator positions are $(c, g)$ with a maximum and minimum of [100, 100] and [0.001, 0.001], respectively.

24. The mean false recognition rate of the training set following three cross-validations constitutes the fitness function. Therefore, the entire optimization process is aimed at finding the global minimum.

25. The predators and prey locations are updated. The process can be divided into three phases:

26. Initial optimization phase ($T < T_{\max}/3$). This phase is the survey phase (i.e., the predator is moving faster than the prey), and the corresponding model can be described as follows:

$$\begin{aligned} \mathbf{M}_i &= \mathbf{R_B} \otimes (\mathbf{E}_i - \mathbf{R_B} \otimes \mathbf{P}_i) \; i = 1, , n \\ \mathbf{P}_i &= \mathbf{P}_i + 0.5\mathbf{R} \otimes \mathbf{M}_i \end{aligned} \tag{12}$$

where $\mathbf{R_B}$ denotes a random vector based on Brownian normal distribution, $\mathbf{E}$ and $\mathbf{P}$ are the optimal predators and prey locations, respectively, $\mathbf{R}$ is a random vector between [0, 1], the symbol $\otimes$ is entry-wise multiplications, $\mathbf{M}$ is step size vector, and $T$ is the current iteration.

i.　Middle optimization phase ($T_{\max}/3 < T < 2T_{\max}/3$). This phase is the coexistence of survey and exploitation (i.e., the predator moves at speed similar to that of the prey), and the corresponding model can be expressed as:

$$\begin{aligned}\mathbf{M}_i &= \mathbf{R_L} \otimes (\mathbf{E}_i - \mathbf{R_L} \otimes \mathbf{P}_i) \; i = 1, \ldots, n/2 \\ \mathbf{P}_i &= \mathbf{P}_i + 0.5\mathbf{R} \otimes \mathbf{M}_i \end{aligned} \tag{13}$$

$$\begin{aligned}\mathbf{M}_i &= \mathbf{R_B} \otimes (\mathbf{R_B} \otimes \mathbf{E}_i - \mathbf{P}_i) \; i = n/2, \ldots, n \\ \mathbf{P}_i &= \mathbf{E}_i + 0.5CF \otimes \mathbf{M}_i \\ CF &= (1 - T/T_{\max})^{(2T/T_{\max})} \end{aligned} \tag{14}$$

where $\mathbf{R_L}$ is a random vector depending on the Lévy normal distribution and $CF$ is an adaptation parameter that regulates the predator step size.

ii.　Post-optimization phase ($T > 2T_{\max}/3$). This phase is the development stage (i.e., the predator moves slower than the prey), and the corresponding model is:

$$\begin{aligned}\mathbf{M}_i &= \mathbf{R_L} \otimes (\mathbf{R_L} \otimes \mathbf{E}_i - \mathbf{P}_i) \; i = 1, \ldots, n \\ \mathbf{P}_i &= \mathbf{E}_i + 0.5CF \otimes \mathbf{M}_i \end{aligned} \tag{15}$$

27. Iterative stagnation due to local optimal points is avoided as follows:

$$\mathbf{P}_i = \begin{cases} \mathbf{P}_i + CF[\mathbf{X}_{\min} + \mathbf{R} \otimes (\mathbf{X}_{\max} - \mathbf{X}_{\min})] \otimes \mathbf{H} \; if \; r \leq F \\ \mathbf{P}_i + [F(1 - r) + r](\mathbf{P}_{r1} - \mathbf{P}_{r2}) \; if \; r > F \end{cases} \tag{16}$$

where $F = 0.2$. Binary vector $\mathbf{H}$ can either be zero or one. $\mathbf{X}_{\max}$ and $\mathbf{X}_{\min}$ are the maximum and minimum values, respectively, $r \in [0, 1]$, and $r_1$ and $r_2$ are random numbers.

28. The prior historical location is replaced if the current prey position's fitness value is lower than its historical value. Otherwise, the historical prey location is retained for the next iteration.

29. The optimization is stopped once the number of iterations is maximal, and the optimization results can be output. Then, the SVM prediction model is built according to the optimization parameters.

30. Intelligent defect diagnosis was achieved by feeding the testing set into the prediction model.

### 3.3. Proposed Fault Diagnosis Scheme for Planetary Gearboxes

Based on the RCMFDE, S-Iso, and MPA-SVM, a novel fault diagnosis method for planetary gearboxes was proposed according to the following steps (Figure 8):

31. Sensors were used to record the vibration signals of the planetary gearbox in different states. A 1:9 ratio exists between training and test samples.

32. The RCMFDE was used to extract features for each group of signals. The parameters of the RCMFDE are set as $N = 3000$, $m = 2$, $c = 6$, $t = 1$, and $s = 25$. Then, 25 entropy features can be obtained for each sample.

33. The dimensionality of the RCMFDE feature set decreased using the S-Iso, and the feature set with low dimensions can be constructed.

34. The training sample feature set and the testing sample feature set are normalized. The MPA-SVM prediction model is constructed using the training set, and this prediction model receives the testing set as input to achieve intelligent fault diagnosis.

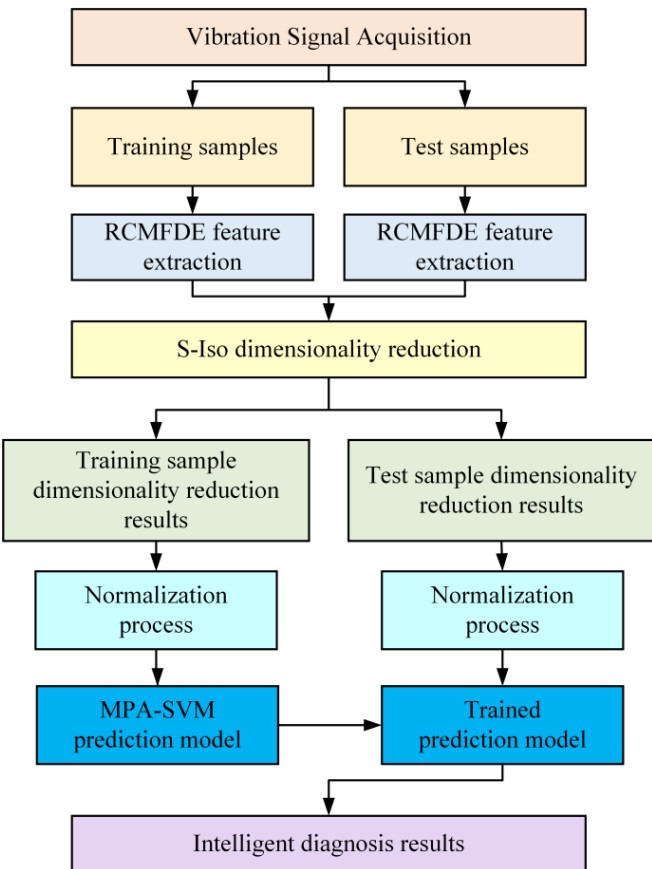

**Figure 8.** Fault diagnosis scheme for planetary gearboxes.

## 4. Planetary Gearbox Fault Diagnosis Experiment

### 4.1. Experimental Platform and Signal Acquisition

The proposed fault diagnosis scheme was utilized to analyze experimental data from a planetary gearbox. Figure 9 depicts the drivetrain dynamics simulator experimental platform, which primarily consists of three-phase asynchronous motor, motor controller, two-stage planetary gearbox, two-stage parallel gearbox, and magnetic powder brake. The number of gear teeth in the planetary gearbox is listed in Table 2.

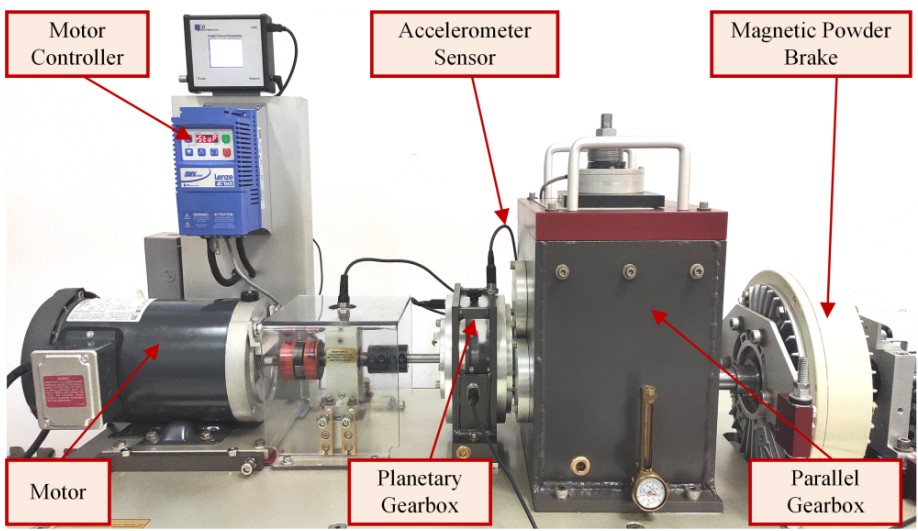

**Figure 9.** Experiment platform.

**Table 2.** The number of gear teeth in the planetary gearbox.

| Component | Number of Gear Teeth | |
|---|---|---|
| | First Stage | Second Stage |
| Gear ring | 100 | 100 |
| Planet gear | 40 (Number of planet gears is 3) | 36 (Number of planet gears is 4) |
| Sun gear | 20 | 28 |

The vibration acceleration signals of the planetary gearbox were acquired using the acceleration sensor, including a single normal state, three bearing fault states, and four gear fault states (Table 3). The waveforms are plotted in Figure 10. For each state, 200 sets of vibration signals were collected at a sample rate of 3000 Hz, and each set of signals contained 3000 sample points. Then, 180 sets were utilized as testing samples, while 20 sets were randomly chosen as training samples. Thus, 1600 sample sets were obtained for eight states of the planetary gearbox, and the total number of training samples and testing samples was 160 and 1440, respectively.

**Table 3.** Fault states of planetary gearboxes.

| Fault State | Abbreviation | Label |
|---|---|---|
| Normal | NOR | 1 |
| Bearing with rolling roller failure | BRF | 2 |
| Bearing with inner ring failure | BIF | 3 |
| Bearing with outer ring failure | BOF | 4 |
| Sun gear with wear failure | GWF | 5 |
| Sun gear with broken tooth failure | GBF | 6 |
| Sun gear with crack failure | GCF | 7 |
| Sun gear with missing tooth failure | GMF | 8 |

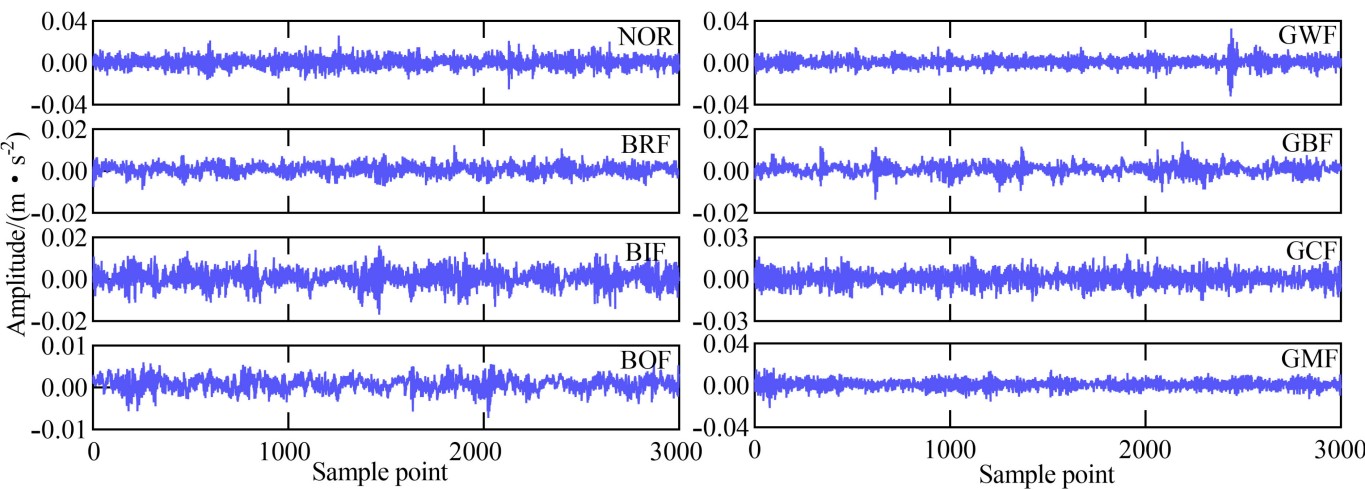

**Figure 10.** Vibration signals of planetary gearboxes.

### 4.2. Fault Feature Extraction

The RCMFDE was first utilized to extract 25 entropy features for each group of samples based on the suggested planetary gearbox fault diagnosis scheme. However, to confirm the benefits of the RCMFDE, the RCMFDE was compared with the MFDE and MPE, and the feature extraction results using three entropy-based methods for planetary gearbox signals are shown in Figure 11.

The following conclusions can be obtained from Figure 11.

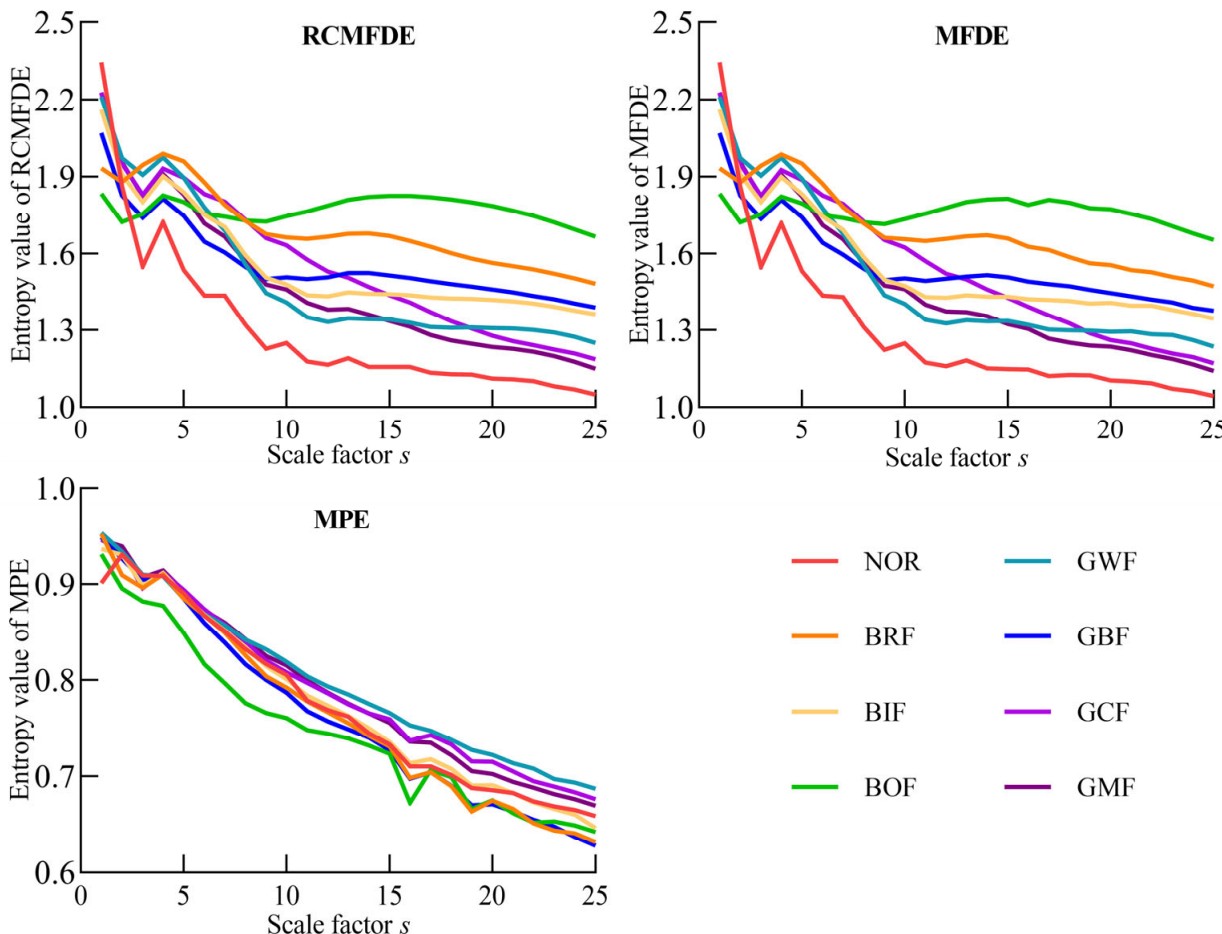

**Figure 11.** Feature extraction results of three methods.

First, when the scale is one, the entropy values of the normal state are higher than that of fault states in the RCMFDE and MFDE feature extraction results, while the entropy values of the normal state are lower than that of fault states in the MPE feature extraction results. For practical purposes, when the planetary gearbox is in a normal condition, the collected vibration acceleration signals are characterized by strong irregularity, increasing the entropy values. Conversely, gathered signals will have regular pulses when the planetary gearbox malfunctions, decreasing entropy values. Therefore, compared with the MPE, feature extraction results of the RCMFDE and MFDE are more realistic.

Second, the mean entropy curves of the MPE for each state are relatively close and difficult to distinguish. At the same time, the RCMFDE and MFDE analysis results clearly show that the planetary gearbox's eight states can be differentiated. This phenomenon confirms that the RCMFDE and MFDE can fully exploit the planetary gearbox fault information.

Finally, compared with the MFDE, the average entropy curves obtained by the RCMFDE are smoother, confirming that the proposed RCMFDE with the refined composite approach can mine fault characteristics more stable.

Furthermore, the MPA-SVM classifier receives the three features in the above sets as input for quantitative analysis, and the diagnostic results are provided in Figure 12.

According to Figure 12, the average recognition rate of the RCMFDE feature set (i.e., 94.72%) is 0.76% and 6.59% higher than that of the MFDE and MPE, respectively, demonstrating the validity of the planetary gearbox feature mining using the RCMFDE. However, since the information redundancy affects the classifier recognition effect, there are 76 samples in the RCMFDE feature set with category misclassification. Therefore, it is essential to make the RCMFDE set less dimensional via the dimensionality reduction technique.

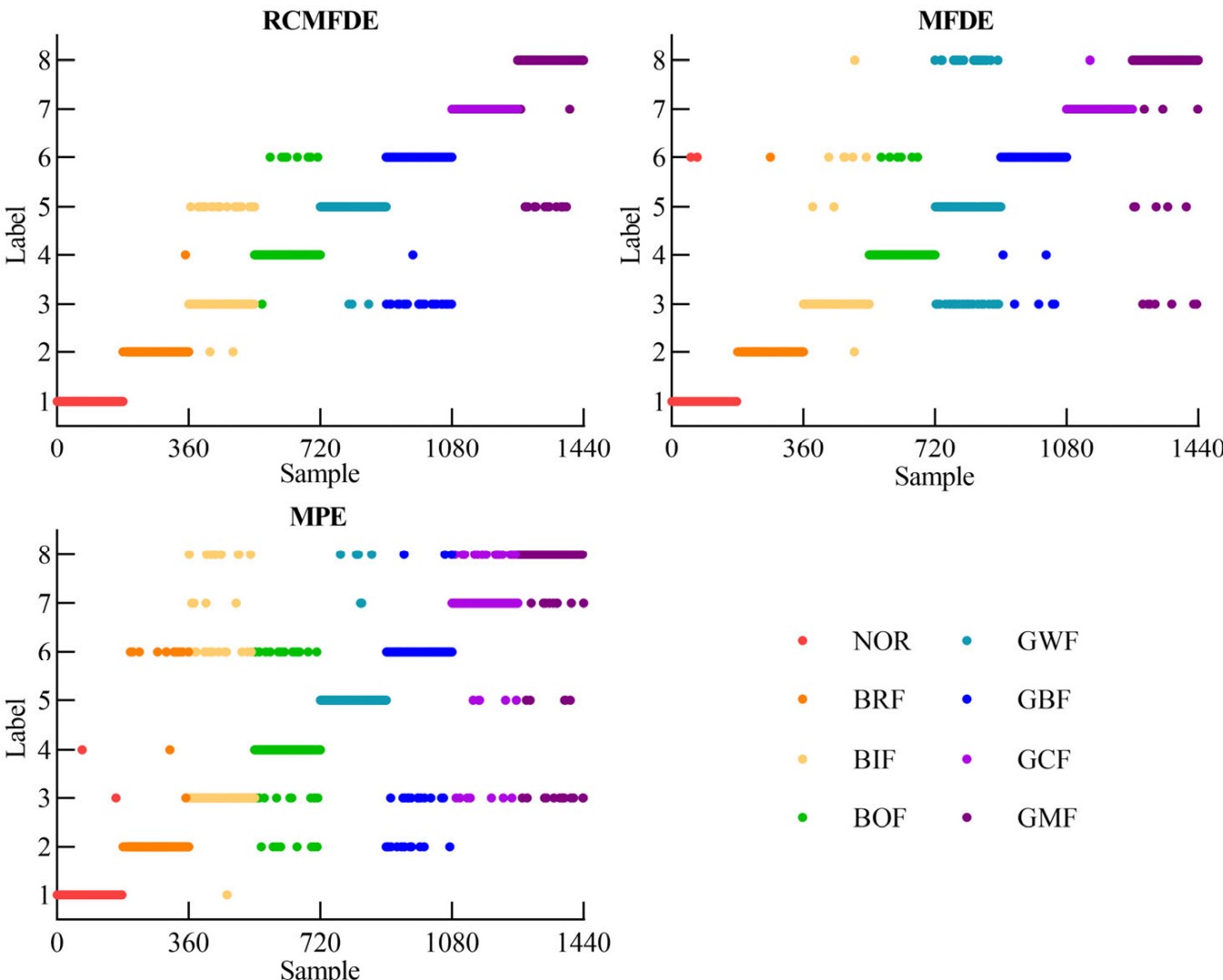

**Figure 12.** Recognition results of three feature sets.

*4.3. Fault Feature Dimensionality Reduction*

According to the suggested method for identifying faults with planetary gearboxes, the S-Iso manifold mapping approach makes the RCMFDE feature set less dimensional. Moreover, the RCMFDE is compared with currently used techniques like PCA, LDA, and Isomap. Figure 13 shows the results of pre-3D visualization. The optimal dimensionality reduction is determined as five using the maximum likelihood estimation, and the remaining parameters are optimally selected by multiple experiments, as listed in Table 4.

According to Figure 13, compared with the LDA and S-Iso, the visualization outcomes of the PCA and Isomap show severe confounding of faulty samples. This is because the unsupervised dimensionality reduction techniques PCA and Isomap do not fully utilize the sample labeling data. Thus, the dimensionality reduction of the PCA and Isomap is not as effective as dimensionality reduction methods LDA and S-Iso. Moreover, compared with the LDA, the S-Iso distinguishes eight states of the planetary gearbox without serious sample overlap, verifying the feasibility of applying the S-Iso to reduce the dimensionality of planetary gearbox feature sets.

In addition, the ratio between the inter-class and intra-class spacing of the dimensionality reduction result was used as the performance index. The quantitative analysis results are shown in Table 5. A higher ratio denotes a more concentrated concentration of related samples and a more evident separation of different samples.

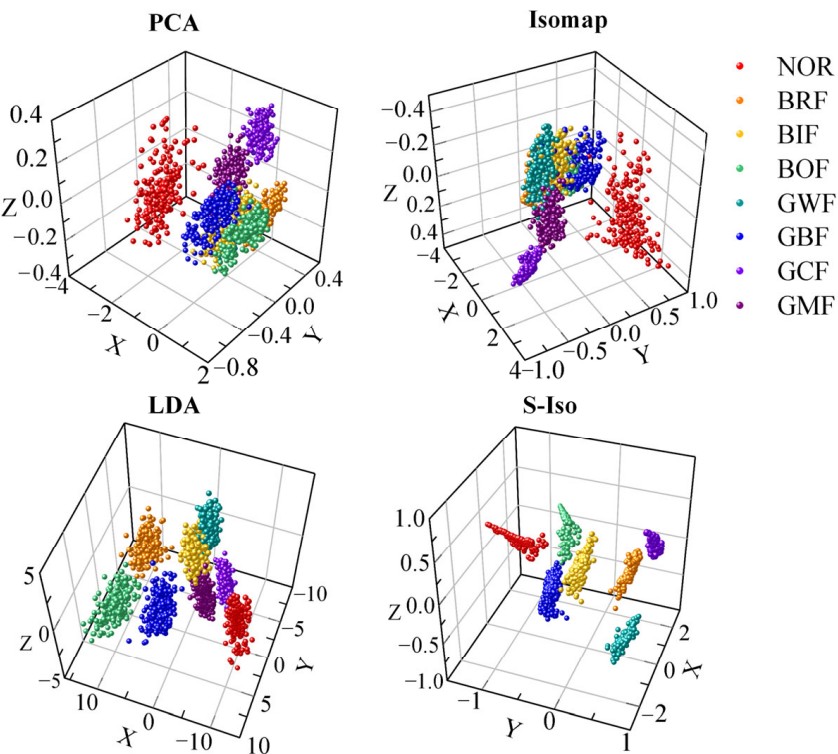

**Figure 13.** Dimensionality reduction results using four methods.

**Table 4.** Parameter settings for different dimensionality reduction methods.

| Method | Parameter Setting |
|--------|-------------------|
| PCA | Dimensionality is 5 |
| LDA | Dimensionality is 5 |
| Isomap | Dimensionality is 5; Nearest neighbor parameter is 38 |
| S-Iso | Dimensionality is 5; Nearest neighbor parameter is 53; and Weighting parameter is 0.4 |

**Table 5.** Performance comparison between four methods.

| Method | Inter-Class Spacing | Intra-Class Spacing | Inter-Class Spacing/ Intra-Class Spacing |
|--------|--------------------|--------------------|------------------------------------------|
| PCA | 0.70 | 0.14 | 5.00 |
| LDA | 58.82 | 4.97 | 11.84 |
| Isomap | 0.99 | 0.18 | 5.50 |
| S-Iso | 1.40 | 0.11 | 12.73 |

Table 5 demonstrates how the linear-based dimensionality reduction approaches (i.e., PCA and LDA) have lower quantization metrics than the nonlinear-based dimensionality reduction methods (i.e., Isomap and S-Iso). This metric can be explained by the fact that the feature set of the planetary gearbox collected using RCMFDE has nonlinear characteristics. Therefore, the nonlinear dimensionality reduction techniques are more appropriate for processing this category of feature set.

Additionally, compared with the unsupervised dimensionality reduction approaches (i.e., PCA and Isomap), supervised dimensionality approaches (i.e., LDA and S-Iso) can significantly improve the performance metrics. This performance can be attributed to the LDA and S-Iso using the sample label information to guide the dimensionality reduction. Therefore, supervised dimensionality reduction methods have better dimensionality reduction effects.

Finally, the S-Iso had the greatest quantitative measurements compared with the three approaches. Hence, it can be concluded that the S-Iso is more suitable for the dimensionality reduction of planetary gearbox feature sets by combining supervised learning with nonlinear manifold learning.

### 4.4. Fault State Identification

According to the suggested planetary gearbox malfunction detection scheme, the MPA-SVM classifier was used to identify faults in the above reduced-dimensional feature sets. The output results are shown in Figure 14.

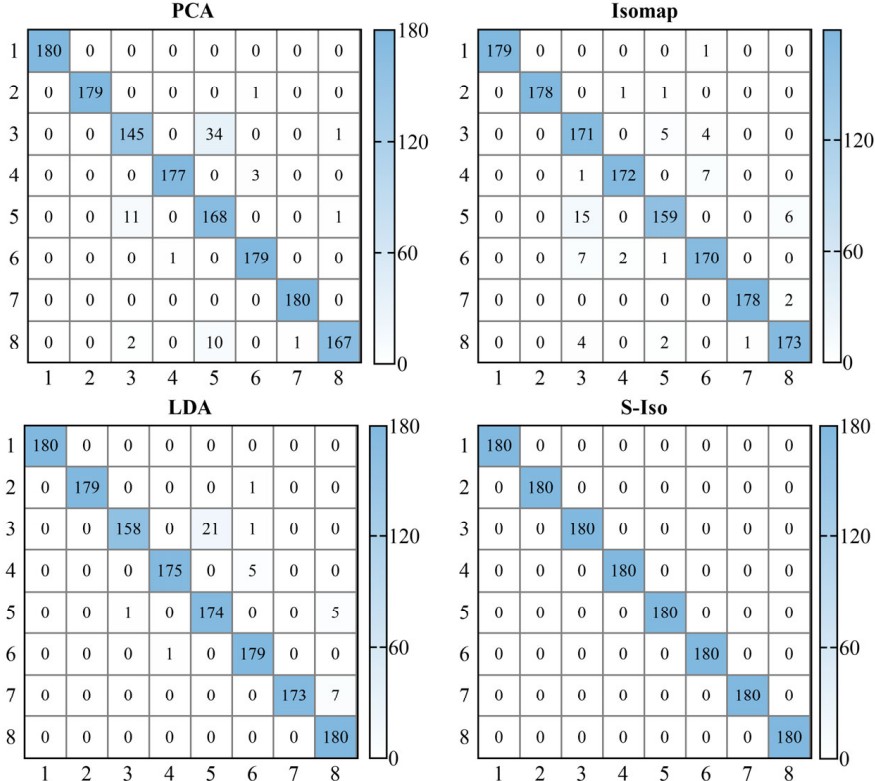

**Figure 14.** Confusion matrices of four methods.

According to Figures 12 and 14, the MPA-SVM has higher recognition accuracy for the reduced-dimensional feature sets compared with the original RCMFDE recognition result. In other words, dimensionality reduction methods can remove redundant information, obtaining the low-dimensional feature set with higher discrimination. Moreover, the averaged recognition rate of the RCMFDE+S-Iso reaches 100%, which is 4.51%, 2.92%, and 4.17% higher than that of the RCMFDE+PCA, RCMFDE+LDA, and RCMFDE+Isomap, respectively. This observation confirms the viability of the feature extraction method that combines the RCMFDE and S-Iso. Finally, the proposed fault diagnosis scheme completely and correctly identifies the labels of 1440 sets of testing samples with 100% recognition accuracy, verifying the viability and superiority of the suggested strategy for the planetary gearbox.

### 5. Conclusions

Based on the refined composite multiscale fluctuation dispersion entropy (RCMFDE), supervised isometric mapping (S-Iso), and marine predators' algorithm-based support vector machine (MPA-SVM), a novel planetary gearbox fault diagnosis scheme was proposed in this paper. Moreover, the planetary gearbox fault diagnosis experiments attest to the efficacy of the suggested approach. The main conclusions of the paper are summarized as follows:

35. The RCMFDE is suitable for extracting fault features of gearboxes. The average recognition rate of the RCMFDE feature set (i.e., 94.72%) is 0.76% and 6.59%, higher than that of the MFDE and MPE, respectively.
36. The S-Iso's visualization effect and performance index are better than the PCA, LDA, and Isomap. Moreover, the averaged recognition rate of the S-Iso reaches 100%, which is 4.51%, 2.92%, and 4.17% higher than that of the PCA, LDA, and Isomap.
37. The suggested fault diagnosis scheme for planetary gearboxes can accurately identify eight states of planetary gearboxes with 100% recognition accuracy.

In this paper, the proposed fault diagnosis scheme can effectively and accurately diagnose gearbox faults. In future work, the authors investigate the parameter settings of the RCMFDE method. Moreover, we will combine the RCMFDE with multivariate technique and apply it to analyze multi-channel signals.

**Author Contributions:** Conceptualization, Z.W.; methodology, H.S. and Z.W.; software, H.S. and Z.W.; validation, H.S. and Y.C.; formal analysis, H.S. and Y.C.; investigation, J.D. and X.W.; resources, L.Y.; data curation, Y.C. and X.W.; writing original draft preparation, H.S. and Z.W.; writing—review and editing, J.D., L.Y. and X.W.; visualization, J.D. and Z.W.; supervision, Z.W.; project administration, Z.W.; funding acquisition, L.Y. All authors have read and agreed to the published version of the manuscript.

**Funding:** This work was supported in part by the National Key R&D Program of China, Grant No. 2022YFB4702401; the National Natural Science Foundation of China, Grant No. 51775114; the Fujian Provincial Science and Technology Major Special Project, Grant No. 2021HZ024006, 2022HZ026025; and the Fujian Provincial High-End Equipment Manufacturing Collaborative Innovation Center, Grant No. 2021-C-275.

**Institutional Review Board Statement:** Not applicable.

**Informed Consent Statement:** Not applicable.

**Data Availability Statement:** Not applicable.

**Acknowledgments:** Not applicable.

**Conflicts of Interest:** The authors declare that they have no known competing financial interests or personal relationships that could have appeared to influence the work reported in this paper.

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
