# Peer review of "Refined Composite Multiscale Fluctuation Dispersion Entropy and Supervised Manifold Mapping for Planetary Gearbox Fault Diagnosis"

_machines, doi:10.3390/machines11010047_

Round 1

Reviewer 1 Report

1) The paper needs to be revised by a native speaker. There are several errors in the text like

such as in line 35 faulty instead of fauly

2) Serious issue is with the word "Improved" before MFDE in the title, Refined composite of coarse grained series is already an established technique. It is hard to understand the claim of improvement, so title of the paper must be modified after removing the the word "Improved" from the title and also from the other sections of the manuscript. 

3) In line 148, 'Five parameters in the RCMFDE should be decided artificially', does not make any sense, so rewrite it with suitable modification.

4) Why blue noise signals are considered for simulation experiments? Why not Pink noise or any other noise series?

5) Conclusion drawn from figure 3 are not appropriate, Statements such as: "The average entropy curves of white noise are larger than those of blue noise" and " Besides, the trends of these two noise entropy curves are obviously different in the whole scale", make no sense, so it is advised to rewrite and modified them.

6) Data in Table 1 is irrelevant, How running time is corelated with the algorithm operating efficiency. The Table 1 doesn't support to set N=3000.

7) Sampling rate is very less to work on 20th scale. Please clarify, How much is the signal length at 20th scale?

Round 2

Reviewer 1 Report

It is recommended to enhance the reference list by adding the references, such as:

Sharma, S. and Tiwari, S.K., 2022. A novel feature extraction method based on weighted multi-scale fluctuation based dispersion entropy and its application to the condition monitoring of rotary machines. Mechanical Systems and Signal Processing171, p.108909.

Reviewer 2 Report

Accept in present form